# Safety and Feasibility of Wedge Resection in Lung Cancer Patients with Pre-Existing Interstitial Lung Disease: Real-World Data from Multicenter, Shizuoka Registry

**DOI:** 10.3390/jcm14165724

**Published:** 2025-08-13

**Authors:** Keigo Sekihara, Kensuke Takei, Koshi Homma, Motohisa Shibata, Kazuhito Funai

**Affiliations:** Division of Surgery First, Hamamatsu University School of Medicine, 1-20-1 Handayama, Chuo Ward, Hamamatsu 431-3192, Japan; k.takei@hama-med.ac.jp (K.T.); koshi@hama-med.ac.jp (K.H.); 41241035@hama-med.ac.jp (M.S.); kfunai@hama-med.ac.jp (K.F.)

**Keywords:** lung cancer, surgery, interstitial lung disease, mortality, morbidity, acute exacerbation, wedge resection, anatomical resection

## Abstract

**Background/Objectives:** Acute exacerbation of interstitial lung disease (AE-ILD) is a life-threatening complication in lung cancer patients with pre-existing ILD. Anatomical resection is recognized as a significant risk factor for AE-ILD. We investigated the safety and feasibility of wedge resection in lung cancer patients with ILD. **Methods:** This retrospective study analyzed clinical stage IA–IIIA primary lung cancer patients with ILD, as recorded in the Shizuoka Registry across eight institutions from January 2019 to May 2023. Patients were categorized into a wedge resection group (WG) and an anatomical resection group (AG), which included segmentectomy, lobectomy, and bilobectomy. Perioperative outcomes were compared between the groups. **Results:** The WG comprised 36 patients, while the AG included 81. The WG had significantly older patients (77 vs. 72 years, *p* < 0.01) and smaller tumors (18 vs. 24 mm, *p* < 0.01). Wedge resection was associated with shorter operative time (100 vs. 205 min, *p* < 0.01) and less blood loss (5 vs. 30 mL, *p* = 0.02). The incidence of postoperative complications did not differ significantly (*p* = 0.84). AE-ILD occurred in three patients (8%) in the WG and four patients (4%) in the AG. Perioperative mortality was 0% in the WG and 2% in the AG; both deaths were due to AE-ILD. Marginal recurrence was observed in four patients (11%) in the WG. **Conclusions:** Although AE-ILD incidence was higher, no deaths due to IP-AE were observed in the WG. While wedge resection cannot completely prevent postoperative AE-ILD, it may reduce perioperative mortality in lung cancer patients with ILD.

## 1. Introduction

The prognosis of lung cancer patients is influenced not only by tumor characteristics but also by comorbidities. Interstitial lung disease (ILD), a representative diffuse parenchymal lung disease characterized by pulmonary fibrosis, is associated with poorer outcomes [1,2]. Idiopathic pulmonary fibrosis (IPF), a common subtype of ILD, can be life-threatening, with a reported median survival time of 2 to 3 years following diagnosis [3,4]. Additionally, previous studies have demonstrated an increased incidence of lung cancer among patients with ILD. Not only respiratory failure due to the progression of ILD itself but also lung cancer are among the leading causes of death in patients with ILD [5]. In addition to its high incidence, the following treatment challenges pose significant difficulties for clinicians.

Surgical resection provides the highest possibility of cure in early-stage lung cancer. However, acute exacerbation of ILD (AE-ILD) is a fatal complication and the leading cause of perioperative mortality in Japanese patients undergoing pulmonary resection [6]. Reports indicate that AE-ILD occurs in 4.2% to 12.4% of cases, with mortality rates as high as 44% [7,8,9]. Although pirfenidone has shown prophylactic efficacy in a prospective trial (WJOG6711L), no standardized preventative strategy for AE-ILD has been established [10]. To reduce postoperative mortality in lung cancer patients with ILD, it is essential to minimize the occurrence of AE-ILD as much as possible. Safe treatment strategies for lung cancer patients with pre-existing ILD is a clinical need.

According to the Japanese Association for Chest Surgery (JACS), seven risk factors for AE-ILD have been identified. Notably, wedge resection is associated with a lower risk of AE-ILD than more extensive resections, such as lobectomy [7]. In high-risk patients, wedge resection is often chosen, with an emphasis on perioperative safety. Based on this, we evaluated the postoperative outcomes of lung cancer patients with ILD to assess the impact of wedge resection in minimizing AE-ILD risk. The objective of this study was to examine the safety and feasibility of wedge resection in lung cancer patients with ILD.

## 2. Materials and Methods

This study was approved by the Institutional Review Board of Hamamatsu University School of Medicine (approval number 24-303). The requirement for informed consent was waived due to the retrospective design.

### 2.1. Shizuoka Registry

We retrospectively reviewed the medical records of patients who underwent operation for diagnosis of primary lung cancer at eight institutions (Hamamatsu University School of Medicine, Seirei Hamamatsu General Hospital, Hamamatsu Medical Center, Iwata city Hospital, Yaizu City Hospital, Fujieda Municipal General Hospital, Japanese Red Cross Shizuoka Hospital, and Fujinomiya City General Hospital) in Shizuoka Prefecture between January 2019 and May 2023. This registry included a total of 1261 patients. Tumors were staged according to the 8th edition of the TNM Classification by the International Union for Cancer Control. Pulmonary function tests were conducted, and the serum Cartino Enbriotic Antigen (CEA) Krebs von den Lungen-6 (KL-6) level was examined preoperatively.

### 2.2. Patient Selection and Diagnosis of Interstitial Lung Disease

We selected patients with ILD who underwent R0 resection for clinical stage IA–IIIA primary lung cancer. R0 resection was defined as complete macroscopic and microscopic tumor removal. Smoking history was defined as habitual smoking for any interval that was carefully evaluated at the preoperative outpatient department and the ward at the time of admission. This study included two surgical approaches, thoracotomy and thoracoscopic surgery. Thoracotomy was performed laterally and assisted by a thoracoscope with a maximum skin incision of 8–10 cm. Four-port thoracoscopic surgery was performed with a maximum skin incision of about 3–4 cm. Patients were categorized into two groups based on the surgical procedure: the wedge resection group (WG) and the anatomical resection group (AG), which included segmentectomy, lobectomy, and bilobectomy. ILD was diagnosed in consultation with internal medicine at each institution based on imaging examination on high-resolution computed tomography (HRCT), pulmonary function tests, and preoperative measurements of the serum KL-6 level. Radiological assessments were reviewed retrospectively and categorized according to the 2018 ATS guidelines [11]. UIP patterns were defined as characterized by a subpleural and basal predominant distribution of honeycomb lesions, which have multiple, similarly sized cystic lesions of 2–10 mm diameter with a thick wall.

### 2.3. Postoperative Acute Exacerbation and Other Complications

Postoperative complications within 30 days of surgery were assessed using the Clavien–Dindo classification [12]; grade III or higher was considered severe. AE-ILD was diagnosed when a patient with ILD acutely developed subjective worsening of dyspnea and met the following criteria: (i) new bilateral radiological opacities on HRCT that were not observed preoperatively; (ii) no evidence of infection, and the possibility of infection was excluded using the laboratory data, radiological findings, and sputum culture; (iii) no evidence of the alternative cause of dyspnea, including left heart failure, pulmonary embolism, or an identifiable cause of acute lung injury; and (iv) a decrease of more than 10 mm Hg in arterial oxygen tension [7,11,13].

### 2.4. Statistical Analyses

Categorical variables were analyzed using Fisher’s exact test and continuous variables using the *t*-test. Kaplan–Meier survival curves were used to estimate overall survival (OS), with group comparisons made using the log-rank test. Univariate and multivariate Cox proportional hazards models were used to identify prognostic factors. OS was defined as the time from surgery to death from any cause or last follow-up. Statistical significance was set at *p* < 0.05. Analyses were conducted using R version 4.3.0 (R Development Core Team 2013, A Language and Environment for Statistical Computing, R Foundation for Statistical Computing, Vienna, Austria. URL: http://www.r-project.org).

## 3. Results

### 3.1. Patient Characteristics

A total of 117 patients with ILD were included in this study. Table 1 summarizes their clinical characteristics.

The incidence of ILD among all patients in the Shizuoka Registry was 11%. The majority were male (n = 73, 92%), with a median age of 73 years (range: 58–86). The median Brinkman index was 1020 (range: 0–4000). The median % vital capacity (%VC) and % diffusing capacity for carbon monoxide (%DLco) were 108% (range: 67–143%) and 84% (range: 42–223%), respectively. The median serum KL-6 level was 574 U/mL. Radiological UIP patterns were observed in 61% of cases. Clinical stages I, II, and III were found in 78%, 18%, and 4% of patients, respectively. Surgical procedures included wedge resection (31%), segmentectomy (8%), lobectomy (60%), and bilobectomy (1%). Surgical approaches included thoracotomy (95%) and thoracoscopic surgery (5%). The median operative time was 171 min (range: 42–553), with a median blood loss of 20 mL (range: 0–789). No significant difference was shown in operation time between surgical approaches (*p* = 0.56). Histologically, 50% had squamous cell carcinoma, 37% adenocarcinoma, 5% large cell neuroendocrine carcinoma (LCNEC), and 8% other types. Pathological UIP patterns were confirmed in 25% of cases.

### 3.2. Comparison Between WG and AG

Table 2 presents the comparison between the WG (n = 37) and AG (n = 82).

The WG included significantly older patients (median age 77 vs. 72 years, *p* < 0.01) and had smaller tumors (17 mm vs. 24 mm, *p* < 0.01). There were no significant differences in smoking history (Brinkman index: 1100 vs. 990; *p* = 0.48), serum KL-6 levels (586 vs. 574 U/mL; *p* = 0.30), radiological UIP patterns (64% vs. 59%; *p* = 0.67), or pulmonary function (%VC: 99% vs. 104%, *p* = 0.08; %DLco: 81% vs. 85%, *p* = 0.86). WG patients were more likely to have clinical stage I disease (97% vs. 70%, *p* < 0.01) and had significantly shorter operative times (99 vs. 203 min, *p* < 0.01) and lower intraoperative blood loss (5 vs. 32 mL, *p* = 0.02). Surgical approaches did not show significant difference, and the majority of surgical approaches were thoracotomies in both groups (92 vs. 96%, *p* = 0.37). Even when analyzed separately by surgical procedure, the operative time was significantly shorter in the WG (*p* < 0.01 in both). The WG had significantly shorter postoperative hospital stay (6 vs. 8 days, *p* = 0.01). The histological distribution and pathological UIP pattern did not significantly differ between the groups. The histological types of two cases recorded as “Others” in Table 1 were mucoepidermoid carcinoma in the WG and large cell lung carcinoma in the AG.

### 3.3. Postoperative Complications

Postoperative complications are shown in Table 3.

Overall, 46 patients (39%) experienced complications—11 (30%) in the WG and 35 (43%) in the AG (*p* = 0.84). No 30-day postoperative deaths occurred in the WG, while two deaths (2%) occurred in the AG, both attributed to AE-ILD. AE-ILD developed in three patients (8%) in the WG and in four (4%) in the AG. Other severe complications were observed only in the AG, including prolonged air leak (n = 5), postoperative bleeding (n = 1), empyema (n = 1), and bronchopleural fistula (n = 1).

### 3.4. Overall Survival and Multivariant Analysis

The median follow-up period was 14 months. One-year overall survival (OS) was significantly lower in the ILD group compared to the non-ILD group (85% vs. 96%, *p* < 0.01) (Figure 1).

Three-year OS was 69% for ILD patients and 90% for non-ILD patients. Univariate analysis showed that ILD was associated with worse OS (HR: 3.70; 95% CI: 2.27–6.02). Among ILD patients with clinical stage I disease, one-year OS did not significantly differ between the WG and AG (96% vs. 79%, *p* = 0.40). Three-year OS rates were 60% (WG) and 68% (AG). The univariate HR for the surgical procedure was 1.53 (95% CI: 0.58–4.06). Margin recurrence occurred in four patients (11%) in the WG during follow-up (Figure 2). Thirteen patients (36%) in the WG developed recurrence. In the AG, nine patients (11%) developed recurrence with significant lower incidence (*p* < 0.01).

Among ILD patients with pathological stage I disease, the analysis showed that one-year OS did not significantly differ between the WG and AG (96% vs. 79%, *p* = 0.40). Three-year OS rates were 82% (WG) and 78% (AG). The univariate HR for the surgical procedure was 1.60 (95% CI: 0.48–5.30, Figure 3). In the pathological stage II, one-year OS did not significantly differ between the WG and AG (80% vs. 77%, *p* = 0.40). Three-year OS rates were 28% (WG) and 67% (AG).

Multivariate analysis identified older age (HR: 1.05; *p* < 0.01), male sex (HR: 2.66; *p* = 0.01), elevated serum CEA (HR: 1.83; *p* = 0.01), advanced pathological stage (HR: 2.19; *p* < 0.01), and non-adenocarcinoma histology (HR: 2.19; *p* < 0.01) as significant unfavorable prognostic factors (Table 4). Although ILD was associated with poorer survival (HR: 1.58), the result did not reach statistical significance (*p* = 0.11).

## 4. Discussion

Balancing perioperative safety with oncological efficacy is critical when determining surgical strategies for lung cancer patients with ILD. In our study, wedge resection was more frequently performed in older patients with smaller tumors. Wedge resection was performed in a minimally invasive manner with shorter operative time (*p* < 0.01) and less blood loss (*p* = 0.02). However, there was no significant difference in the frequency of AE-ILD (8% vs. 4%, *p* = 0.68). Despite being less invasive, wedge resection showed comparable postoperative complication rates to anatomical resection, and notably, no perioperative mortality was observed in the wedge group.

AE-ILD remains one of the most serious postoperative complications following lung resection in ILD patients [7]. Despite efforts to develop prophylactic treatments, such as perioperative use of pirfenidone, AE-ILD continues to pose significant risks [6,14]. In our cohort, AE-ILD occurred in 6% of ILD patients, with a 29% mortality rate among affected individuals. This highlights the importance of cautious treatment planning in this population. Previous studies have also demonstrated that AE can be triggered by other lung cancer treatments, such as chemotherapy and radiotherapy [15,16,17]. Immune checkpoint inhibitors and carbon-ion radiotherapy are among the latest treatment modalities being explored to establish safe therapeutic strategies for patients with lung cancer complicated by interstitial pneumonia [18,19,20,21]. Even among these various treatment modalities, given the unpredictable nature of AE-ILD, surgical decision-making must weigh curability against risk.

Recently, treatment for NSCLC has been improved in stereotactic body radiotherapy (SBRT) and molecular target drugs. SBRT was administered to 55 patients with stage I NSCLC who were deemed operable but declined surgery. The 3-year OS rate was 82% [22]. However, another report demonstrated that the fetal AE-ILD rate was 16.7%, significantly higher than that observed in the non-ILD group (0.8%) [23]. Osimertinib and Alectinib have demonstrated excellent outcomes as adjuvant therapies and have transformed the standard treatment of NSCLC [24,25,26]. However, according to real-world clinical data, 6.5–6.8% of 3578 patients treated with Osimertinib developed ILD, and 29 of them experienced fatal outcomes. In patients with pre-existing ILD, the risk of ILD onset increases by approximately 3.5-fold [27]. As a result, these populations have been excluded from clinical trials of these agents. Patients with ILD are not eligible to benefit from these novel treatments. The development of safe treatment strategies for lung cancer patients with comorbid ILD is an urgent need.

The choice of surgical procedure is important. The large-scale prospective study of the postoperative AE-ILD by the JACS demonstrated seven risk factors: (i) male sex, (ii) preoperative steroid use for ILD, (iii) previous history of AE, (iv) radiological UIP pattern, (v) %VC less than 80%, (vi) elevation of the serum KL-6 level, and (vii) surgical procedure besides segmentectomy [7]. Consequently, wedge resection is often considered a safer surgical option in this context. In our study, the wedge resection rate was 31%, nearly double that of previous reports [1]. This patient selection strategy contributed to a lower observed incidence of AE-ILD and achieved zero perioperative mortality in the WG. However, the incidence of AE-ILD was not significantly different between the WG and AG (8% vs. 4%, *p* = 0.68), indicating that wedge resection cannot entirely eliminate the risk.

Previous studies of surgical biopsy for ILD diagnosis have similarly reported the safety of wedge resection, with extremely low morbidity and mortality [28,29]. However, therapeutic resections for malignancy involve additional risks, especially in patients with high-risk profiles. The high proportion (61%) of patients with a radiological UIP pattern in our study may explain the persistence of AE-ILD despite conservative surgical approaches.

Multivariate analysis showed that male sex (HR 2.66), advanced pathological stage (HR 2.19), and non-adenocarcinoma histology (HR 2.19) have a strong influence among the significant independent prognostic factors. The ILD group had high frequencies of these unfavorable characteristics—92% were male, 34% had advanced-stage cancer, and 50% had squamous cell carcinoma. Consequently, ILD was not independently associated with poor overall survival in multivariate analysis. In the short term, wedge resection showed a slight advantage in 1-year OS, likely reflecting its lower perioperative risk. Although wedge resection may be acceptable from a safety standpoint in patients with ILD, it has not been adopted as the standard surgical procedure due to lower survival rates caused by a high incidence of local recurrence, including recurrence at the surgical margins [30]. In our study, 11% of patients in the WG developed local recurrence at the surgical margin. This may explain the convergence of long-term survival curves between the WG and AG. Similar trends have been reported previously, with lobectomy demonstrating superior survival in some cohorts [14]. While long-term follow-up is necessary to draw definitive conclusions, recent evidence supports the efficacy of sublobar resection for small, peripheral lung cancers [31]. The median tumor size in our study (17 mm) aligns with these indications, suggesting that wedge resection may offer comparable long-term outcomes if appropriately selected.

This study has several limitations. First, it is a retrospective analysis across multiple institutions, and the follow-up period was relatively short (median: 14 months). And surgical protocols across institutions could introduce variability. Second, the total number of objective patients was small, with no mention of propensity score matching or adjusted multivariate models comparing the WG vs. AG. We consider any adjustments or statistical controls important. We will continue accumulating cases and address them as a subject for analysis in the future. Third, the classification of ILD patterns was based on radiological findings, which are subject to interobserver variability. Although thoracic surgeons reviewed imaging in consultation with radiologists and pulmonologists, some diagnostic inconsistency may have occurred. Fourth, while recent guidelines (2022) offer updated criteria, our study relied on the 2018 ATS guideline, which was applicable to the majority of our cohort.

In conclusion, wedge resection can reduce perioperative mortality in lung cancer patients with ILD but does not eliminate the risk of AE-ILD. Moreover, it may not offer superior long-term survival. Careful patient selection and long-term monitoring remain essential.

## 5. Conclusions

Wedge resection appears to be a safer surgical option for lung cancer patients with underlying interstitial lung disease, significantly reducing perioperative mortality compared to anatomical resection. However, it does not eliminate the risk of postoperative acute exacerbation of ILD or show significant difference in mortality. Additionally, wedge resection may offer limited benefit in terms of long-term survival. Therefore, surgical strategies for this population must be individualized, taking into account tumor characteristics, patient comorbidities, and the risks of postoperative complications. Continued prospective studies and long-term follow-up are essential to better defining the role of wedge resection in this high-risk patient group.

## Figures and Tables

**Figure 1 jcm-14-05724-f001:**
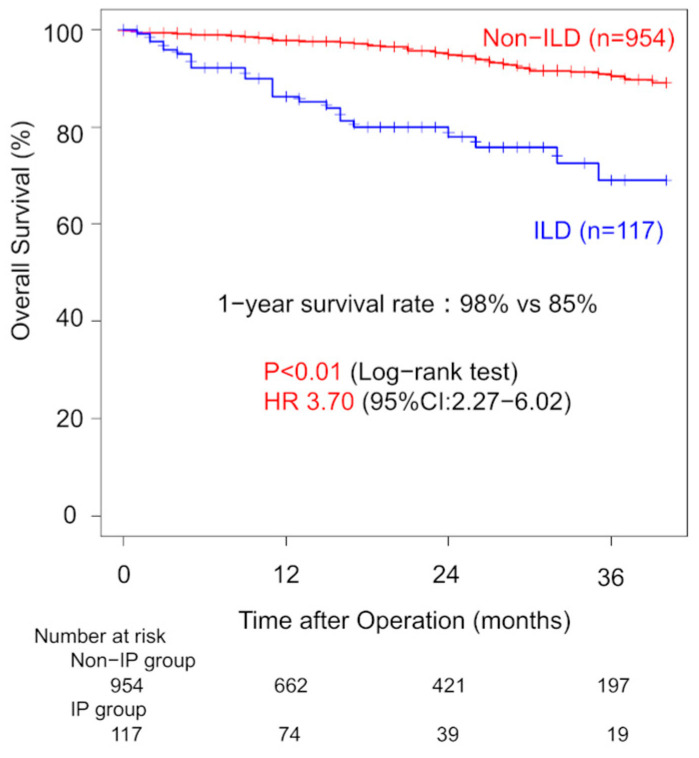
Comparison of overall survival between patients with and without interstitial lung disease.

**Figure 2 jcm-14-05724-f002:**
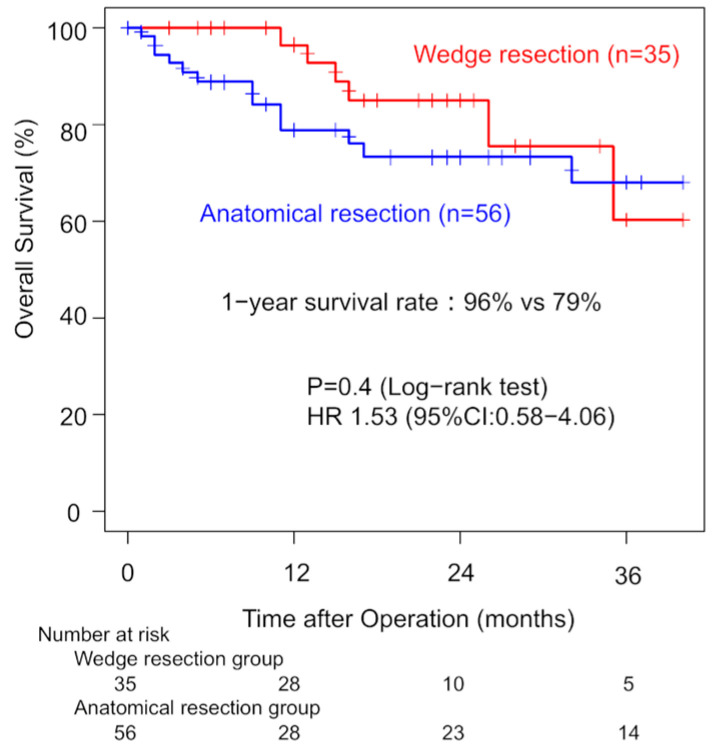
Comparison of overall survival in clinical stage I patients between the wedge resection group and the anatomical resection group.

**Figure 3 jcm-14-05724-f003:**
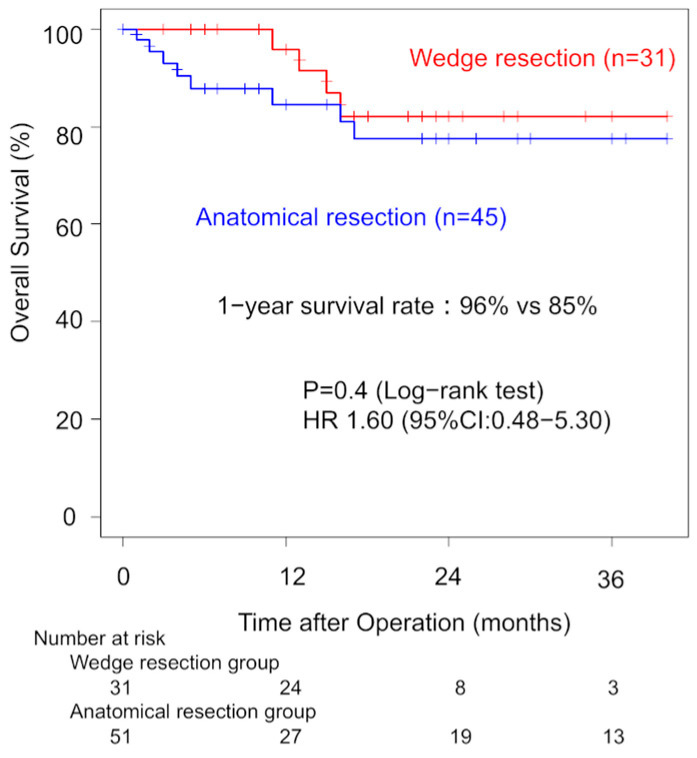
Comparison of overall survival in pathological stage I patients between the wedge resection group and the anatomical resection group.

**Table 1 jcm-14-05724-t001:** Clinicopathological characteristics of patients with interstitial lung disease.

		Patients with ILD (n = 117)
Age		73 (58–86)
Sex (%)	Male	108 (92)
	Female	9 (8)
Smoking history (%)	Presence	116 (99)
Brinkmann index		1000 (21–4000)
KL-6		574 (210–2044)
CEA		4.9 (0.9–71)
Respiratory function	%VC	102 (67–143)
	FEV1.0%	74 (51–120)
	%DLCO	84 (42–223)
Comorbidity (%)	ACS	15 (13)
	DM	33 (29)
	Collagen disease	19 (16)
Radiological tumor size (mm)		22 (5–112)
Clinical stage (%)	I	91 (78)
	II	21 (18)
	III	5 (4)
Radiological UIP pattern (%)		71 (61)
Surgical procedure (%)	Wedge resection	36 (31)
	Segmemtectomy	10 (8)
	Lobectomy	70 (60)
	Bilobectomy	1 (1)
Surgical approach (%)	Thoracotomy	111 (95)
	Thoracoscopic surgery	6 (5)
Operation time (minute)	Total	171 (42–553)
	Thoracotomy	171 (54–553)
	Thoracoscopic surgery	172 (42–265)
Blood loss (mL)		20 (0–789)
Pathological stage (%)	I	76 (66)
	II	28 (24)
	III	13 (11)
Pathological tumor size (mm)		27 (3–120)
Histology	Sq	58 (50)
	Ad	45 (37)
	LCNEC	6 (5)
	Small	3 (3)
	Sarcomatoid	3 (3)
	Others	2 (2)
Pathological UIP pattern (%)		26 (25)

**Table 2 jcm-14-05724-t002:** Clinicopathological characteristics of wedge resection group and anatomical resection group.

		WG (n = 36)	AG (n = 81)	*p* Value
Age		77(58–86)	72 (60–84)	<0.01
Sex (%)	Male	35 (97)	73 (90)	0.27
	Female	1 (3)	8 (10)	
Smoking history (%)	Presence	37 (100)	80 (99)	1.0
Brinkmann index		1090 (150–3552)	980 (0–4000)	0.48
KL-6		586 (212–1967)	574 (20–2044)	0.30
CEA		5.0 (0.9–70.5)	5.0 (0.9–51.9)	0.28
Respiratory function	%VC	101 (67–134)	104 (69–143)	0.08
	FEV1.0%	73 (51–91)	74 (53–120)	0.25
	%DLCO	81 (44–223)	85 (42–142)	0.86
Comorbidity (%)	ACS	7 (19)	8 (10)	0.23
	DM	11 (30)	22 (27)	0.82
	Collagen disease	7 (19)	12 (15)	0.59
Radiological tumor size (mm)		18 (10–48)	24 (5–112)	<0.01
Clinical stage (%)	I	35 (97)	56 (70)	<0.01
	II	1 (3)	20 (24)	
	III	0 (0)	5 (6)	
Radiological UIP pattern (%)		23 (64)	48 (59)	0.67
Surgical procedure (%)	Wedge resection	36 (100)	-	
	Segmemtectomy	-	10 (12)	
	Lobectomy	-	70 (87)	
	Bilobectomy	-	1 (1)	
Surgical approach (%)	Thoracotomy	33 (92)	78 (96)	0.37
	Thoracoscopic surgery	3 (8)	3 (4)	
Operation time (minute)	Total	100 (42–413)	205 (54–553)	<0.01
	Thoracotomy	100 (58–413)	200 (54–553)	<0.01
	Thoracoscopic surgery	93 (42–110)	247 (234–265)	<0.01
Postoperative hospital stay (day)		6 (2–23)	8 (3–54)	0.01
Blood loss (mL)		5 (0–275)	30 (0–789)	0.02
Pathological stage (%)	I	31 (86)	45 (56)	<0.01
	II	5 (14)	23 (28)	
	III	0 (0)	13 (16)	
Pathological tumor size (mm)		22 (3–120)	30 (3–120)	0.43
Histology	Sq	17 (47)	42 (52)	0.84
	Ad	15 (41)	30 (37)	1.0
	LCNEC	1 (3)	5 (6)	
	Small	1 (3)	2 (2)	
	Sarcomatoid	1 (3)	2 (2)	
	Mucoepidermoid	1 (3)	0 (0)	
	Large	0 (0)	1 (1)	
Pathological UIP pattern (%)		7 (25)	19 (25)	1.0

**Table 3 jcm-14-05724-t003:** The comparison of postoperative complication and mortality between the wedge resection group and anatomical resection group.

			WG (n = 36)	AG (n = 81)	*p* Value
Complication (%)			11 (30)	35 (43)	0.84
	AE-IP	Gr.4	3 (8)	2 (2)	0.67
		Gr.5	-	2 (2)	
	Bleeding	Gr.3b	-	1 (1)	
	Prolonged	Gr.3a	1 (3)	11 (13)	
	air leakage	Gr.3b	-	5 (6)	
	Pleurisy	Gr.2	4 (11)	-	
		Gr.3a	1 (3)	1 (1)	
	Empyema	Gr.3a	-	1 (1)	
		Gr.3b	-	1 (1)	
	BPF	Gr.3b	-	1 (1)	
	Pneumonia	Gr.2	-	3 (4)	
	Others	Gr.2	2 (5)	7 (8)	
Severe complication (%)			3 (8)	12 (15)	0.39
Mortality (%)			0 (0)	2 (2)	1.0

**Table 4 jcm-14-05724-t004:** Multivariate analysis for overall survival.

		Number of Patients	Hazard Ratio (95% CI)	*p* Value
Age		-	1.05 (1.01–1.09)	<0.01
Sex	Female	387	Ref	
	Male	664	2.66 (1.24–5.68)	0.01
Cigarette smoking	Absent	336	Ref	
	Present	715	0.75 (0.34–1.68)	0.49
Serum CEA level	<5.0	815	Ref	
	>5.0	236	1.83 (1.14–2.94)	0.01
Surgical	Wedge resection	127	Ref	
procedure	Anatomical resection	924	0.93 (0.47–1.87)	0.85
Pathological stage	0 or I	820	Ref	
	II, III, IV	231	2.19 (1.37–3.51)	<0.01
Histology	Adenocarcinoma	810	Ref	
	Non-adenocarcinoma	241	2.19 (1.37–3.51)	<0.01
Interstitial lung	Absent	935	Ref	
disease	Present	117	1.58 (0.90–2.80)	0.11

## Data Availability

The data that support the findings of this study are available from the corresponding author upon reasonable request.

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
