# Peer review of "Safety and Feasibility of Wedge Resection in Lung Cancer Patients with Pre-Existing Interstitial Lung Disease: Real-World Data from Multicenter, Shizuoka Registry"

_jcm, 2025, doi:10.3390/jcm14165724_

Round 1

Reviewer 1 Report

Comments and Suggestions for Authors

In the manuscript, the authors present a study regarding the feasibility of the wedge resections in patients with lung cancer and previous interstitial lung disease. It is a multicenter study.  they included in the study 117 patients that were divided in 2 groups (wedge resection and anatomical resections). In my opinion, it is an interesting manuscript. In order to improve the quality of the manuscript, some changes have to be done. My observations are : 

  • in the results section of the abstract, the authors stated that they included  in the study 37 patients in the wedge resections group and 82 patients in the anatomical resections group. In the table 1, the authors registered 117 patients. Something is not correct here. 37+82=119. So, I think the statistical analysis should be corrected.
  • It is not clear in the manuscript what surgical approach what used in this cases. Please include some data regarding the surgical approach that was used in this cases (open, VATS, RATS). I think that the operative time should be study based on the surgical approach that was used.
  • please include some data regarding the period of hospitalization recorded in this cases (wedge resections versus anatomical resections)
  • i think that the 3 years survival rates should be presented based on the stages of the tumors and the type of the surgical procedure that was used not only based on the surgical procedure that was used.
  • the authors recorder 2 cases of "other" type of histopathological type of tumor. Please present what kind of histopathological type of tumor was recorded. 

Author Response

Comments and Suggestions for Authors

Reviewer #1: In the manuscript, the authors present a study regarding the feasibility of the wedge resections in patients with lung cancer and previous interstitial lung disease. It is a multicenter study.  they included in the study 117 patients that were divided in 2 groups (wedge resection and anatomical resections). In my opinion, it is an interesting manuscript. In order to improve the quality of the manuscript, some changes have to be done.

Comment #1: In the results section of the abstract, the authors stated that they included  in the study 37 patients in the wedge resections group and 82 patients in the anatomical resections group. In the table 1, the authors registered 117 patients. Something is not correct here. 37+82=119. So, I think the statistical analysis should be corrected.

Answer #1: Thank you for your careful reading. The total number was “117”. We revised the total number in top rows of Table1 and 2. In the survival analysis and multivariate analysis, we used correct number. We only made a mistake in the Tables.

Comment #2: It is not clear in the manuscript what surgical approach what used in these cases. Please include some data regarding the surgical approach that was used in these cases (open, VATS, RATS). I think that the operative time should be study based on the surgical approach that was used. Please include some data regarding the period of hospitalization recorded in these cases (wedge resections versus anatomical resections).

Answer #2: Thank you for your thoughtful comment. Based on your advice, we added additional analyses and added the results both in Table1 and 2. And we have also described it in the main text as below.

Changes #2: (Result, line 147-151) Surgical approach did not show significant difference, majority of surgical approach were thoracotomy both in the two groups (92 vs 96%, p=0.37). Even when analyzed separately by surgical procedure, the operative time was significantly shorter in the WG (p<0.01 in both). WG had significantly shorter postoperative hospital stay (6 vs. 8 days, p=0.01).

Comment #3: I think that the 3 years survival rates should be presented based on the stages of the tumors and the type of the surgical procedure that was used not only based on the surgical procedure that was used.

Answer #3: Thank you for your thoughtful comment. We added the results of survival analysis depended on the pathological stage. However, pStageII included only 5 patients in the WG, the small sample size resulted in a jagged survival curve, making it unsuitable for publication. We described the result in the text only.

Changes #3: (Result, line182-189) Among ILD patients with pathological stage I disease, the analysis showed one-year OS did not significantly differ between the WG and AG (96% vs. 79%, p=0.40). Three-year OS rates were 82% (WG) and 78% (AG). The univariate HR for surgical procedure was 1.60 (95% CI: 0.48–5.30, Figure 3). In the pathological stage II, one-year OS did not signifi-cantly differ between the WG and AG (80% vs. 77%, p=0.40). Three-year OS rates were 28% (WG) and 67% (AG). Figure 3: Comparison of overall survival in pathological stage I patients between the wedge resection group and the ana-tomical resection group.

Comment #4: The authors recorder 2 cases of "other" type of histopathological type of tumor. Please present what kind of histopathological type of tumor was recorded.

Answer #4: Thank you for your careful reading. The “Others” included mucoepidermoid carcinoma in WG and large cell lung carcinoma in AG. We have described these histological types in the main text as below.

Changes #4: (Result, line 161-163)The histological type of 2 cases recorded as “Others” in Table 1 were mucoepidermoid carcinoma in WG and large cell lung carcinoma in AG.

Reviewer 2 Report

Comments and Suggestions for Authors

Thank you for the opportunity to review your manuscript on the safety and feasibility of wedge resection in lung cancer patients with interstitial lung disease. This is an important and timely topic with strong clinical relevance. Your use of multicenter real-world data and focus on a high-risk patient population are commendable.

Please find my detailed review comments attached, which include suggestions for clarifying the methodology, addressing potential confounders, and refining the interpretation of your results. I hope these comments are helpful in strengthening the clarity, rigor, and overall impact of your study.

Best regards,

----------------------------------

Peer Review Report

Title of Manuscript:
Safety and Feasibility of Wedge Resection in Lung Cancer Patients with Pre-existing Interstitial Lung Disease; Real-World Data from multicenter, Shizuoka registry

Abstract

Clarify Study Objective:
The background mentions “efficacy,” but the study focuses more on safety and feasibility (mortality/morbidity). Replacing “efficacy” with “safety and feasibility” would align better with the presented results. (lines 24).

Baseline differences (age, tumor size) between WG and AG may confound results. The abstract does not mention any adjustments or statistical controls (e.g., multivariate analysis, propensity score matching) for these differences. This weakens the comparison.

The incidence of AE-ILD appears slightly higher in the wedge group (8% vs 4%), yet wedge resection is proposed as safer. While mortality was lower in WG, the abstract should briefly clarify this apparent contradiction (e.g., “although AE-ILD incidence was higher, no AE-ILD-related deaths occurred in the WG”).

Conclusion Needs More Nuance: The conclusion slightly overstates the benefit. Saying wedge resection “may reduce perioperative mortality” is fair, but noting the non-zero AE-ILD incidence and lack of statistical significance in mortality would improve precision.

In the introduction part;

Clarify the study’s objective (i.e., focus on perioperative safety and long-term outcomes).

Minor grammatical corrections would improve flow (e.g., “prognosis is influenced not only by tumor characteristics but also by comorbidities”).

Methodology

Well-defined patient selection criteria. But, Patients were not randomized and wedge resection was performed in older patients with smaller tumors—this introduces a significant confounder. No mention of propensity score matching or adjusted multivariate models comparing WG vs AG, which limits the reliability of group comparisons.

Median follow-up of 14 months is short for evaluating survival and recurrence, especially in early-stage lung cancer.

Results

AE-ILD rates were higher in WG (8%) vs AG (4%), yet wedge resection is described as safer. This discrepancy is explained later but should be acknowledged earlier and explicitly discussed.

  • Include p-values for AE-ILD and recurrence rate differences.
  • Local recurrence in WG (11%) is significant and should be better emphasized and contextualized.

Discussion and Interpretation

  1. The claim that wedge resection reduces mortality is supported, but conclusions on oncological equivalence are limited due to local recurrence and short follow-up.
  2. Discuss other possible treatments (e.g., SBRT, targeted therapy) for small tumors in ILD patients, especially in high-risk surgical candidates.
  3. Address the clinical relevance of margin recurrence more directly.
  4. Highlight that the 8% AE-ILD rate in WG suggests risk is not eliminated, which should caution against overreliance on wedge resection.

Conclusion

Add that the lack of standardized surgical protocols across institutions could introduce variability. (Suggestion)

Improve grammar and fluency in places (e.g., "The ILD group had high frequencies of these unfavorable characteristics” → “The ILD group showed a high prevalence of unfavorable characteristics…”).

Author Response

Reviewer #2:

Abstract

Comment #1: The background mentions “efficacy,” but the study focuses more on safety and feasibility (mortality/morbidity). Replacing “efficacy” with “safety and feasibility” would align better with the presented results. (lines 24).

Answer #1: Thank you for your reading and thoughtful comment. We revised the sentence as your advice.

Comment #2: Baseline differences (age, tumor size) between WG and AG may confound results. The abstract does not mention any adjustments or statistical controls (e.g., multivariate analysis, propensity score matching) for these differences. This weakens the comparison.

Answer #2: We could not identify both ILD and Surgical procedure in the multivariant analysis. Then, we did not describe in the abstract. As your comment, we also consider to mention any adjustments or statistical controls important. We performed a multivariate analysis; however, due to the word limit of the abstract, the results are only described in the Results section.

Comment #3: The incidence of AE-ILD appears slightly higher in the wedge group (8% vs 4%), yet wedge resection is proposed as safer. While mortality was lower in WG, the abstract should briefly clarify this apparent contradiction (e.g., “although AE-ILD incidence was higher, no AE-ILD-related deaths occurred in the WG”).

Answer #3: We added the sentence as your advice, in conclusion section of the abstract.

Comment #4: Conclusion Needs More Nuance: The conclusion slightly overstates the benefit. Saying wedge resection “may reduce perioperative mortality” is fair, but noting the non-zero AE-ILD incidence and lack of statistical significance in mortality would improve precision.

Answer #4: We already described the non-zero AE-ILD incidence as “While wedge resection cannot completely prevent postoperative AE-ILD.”. And as your comment #3, we added the sentence “Although AE-ILD incidence was higher, no deaths due to IP-AE were observed in the WG.”. We could improve the conclusion of abstract, thanks to your advice.

Introduction

Comment #5: Clarify the study’s objective (i.e., focus on perioperative safety and long-term outcomes).

Answer #5: Thank you for your thoughtful comment, we added the sentence about the object in Introduction section.

Changes #5: (Introduction, line 60-61): The object of this study was to examine the safety and feasibility of wedge resection in lung cancer patients with ILD.

Comment #6: Minor grammatical corrections would improve flow (e.g., “prognosis is influenced not only by tumor characteristics but also by comorbidities”).

Answer #6: Thank you for your reading. We have already revised the sentence.

Methodology

Comment #7: Well-defined patient selection criteria. But, Patients were not randomized and wedge resection was performed in older patients with smaller tumors—this introduces a significant confounder. No mention of propensity score matching or adjusted multivariate models comparing WG vs AG, which limits the reliability of group comparisons.

Answer #7: As your comment and we already provided our response in the Comment #2. We also consider to mention any adjustments or statistical controls important. And we performed multivariant analysis. In the OS, WG included only 6 events. So, propensity score matching or adjusted multivariate models was not inappropriate for this study. We added the sentence about this limitation as below.

Changes #7: (Discussion, line 269-273) Second, the total number of objective patients was small, no mention of propensity score matching or adjusted multivariate models comparing WG vs AG. We consider to mention any adjustments or statistical controls important. We will continue accumulating cases and address it as a subject for analysis in the future.

Comment #8: Median follow-up of 14 months is short for evaluating survival and recurrence, especially in early-stage lung cancer.

Answer #8: Thank you for your thoughtful comment. We also think follow up term was not sufficient, then we have already described in the first limitation (line ****).

Results

Comment #9: AE-ILD rates were higher in WG (8%) vs AG (4%), yet wedge resection is described as safer. This discrepancy is explained later but should be acknowledged earlier and explicitly discussed.

Answer #9: As your comment, it was main result of this study, thank you for your understanding. However, the Results section already has a well-structured flow, and it is not feasible to place only this particular result at the beginning. Since it is already mentioned in both the Abstract and the first paragraph of the Discussion, we considered that to be sufficient.

Comment #10: Include p-values for AE-ILD and recurrence rate differences.

Answer #10: About AE-ILD, we have already described, the p-value was 0.67. According to recurrence, we added the results.

Changes #10: (Result, line176-178) Thirteen patients (36%) in the WG developed recurrence. In the AG, 9 patients (11%) developed recurrence with significant lower incidence (p<0.01).

Comment #11: Local recurrence in WG (11%) is significant and should be better emphasized and contextualized.

Answer #11: We also the result was important. We revised the expression to "local recurrence" and added the statement in the abstract.

Changes #11: (Abstract, line 34-35) Marginal recurrence was observed in 4 patients (11%) in WG.

Discussion and Interpretation

Comment #12: The claim that wedge resection reduces mortality is supported, but conclusions on oncological equivalence are limited due to local recurrence and short follow-up.

 Answer #12: Thank you for your thoughtful comment, similar comments were made in other reviews as well. We added the sentence in Conclusion section as below.

Changes #12: (Conclusion, line 287-288) Additionally, wedge resection may offer limited benefit in terms of long-term survival.

 Comment #13: Discuss other possible treatments (e.g., SBRT, targeted therapy) for small tumors in ILD patients, especially in high-risk surgical candidates.

Answer #13: We added the paragraph and reference about SBRT and molecular target therapy in Discussion as below.

Changes #13: (Discussion, line 219-231) Recently, treatment for NSCLC has been improved in the Stereotactic body radio-therapy (SBRT) and molecular target drug. SBRT was administered to 55 patients with stage I NSCLC who were deemed operable but declined surgery. The 3-year OS rate was 82% [23]. Although another report demonstrated that the fetal AE-ILD rate was 16.7%, significantly higher than that observed in the non-ILD group (0.8%) [24]. Osimertinib and Alectinib have demonstrated excellent outcomes as adjuvant therapies and have trans-formed the standard treatment of NSCLC [25-27]. However, according to real-world clini-cal data, 6.5–6.8% of 3,578 patients treated with Osimertinib developed ILD, and 29 of them experienced fatal outcomes. In patients with pre-existing ILD, the risk of ILD onset increases by approximately 3.5-fold [28]. As a result, these populations have been exclud-ed from clinical trials of these agents. Patients with ILD are not eligible to benefit from these novel treatments. The development of safe treatment strategies for lung cancer pa-tients with comorbid ILD is an urgent need.

(Reference)

  1. Kocak Uzel E, Bagci Kilic M, Morcali H, Uzel O. Stereotactic body radiation therapy for stage I medically operable non-small cell lung cancer. Sci Rep. 2023;13:10384.
  2. Ueki N, Matsuo K, Nishii Y, Nishimura Y, Kanayama M, Okamoto I, et al. Outcomes of stereotactic body radiotherapy for stage I non-small cell lung cancer in patients who are medically operable but refuse surgery. J Thorac Oncol. 2015;10:826–31.
  3. Wu YL, Tsuboi M, He J, John T, Grohe C, Majem M, et al. Osimertinib in resected EGFR-mutated non-small-cell lung cancer. N Engl J Med. 2020;383:1711–23.
  4. Tsuboi M, Herbst RS, John T, Kato T, Majem M, Grohé C, et al. Overall survival with osimertinib in resected EGFR-mutated NSCLC. N Engl J Med. 2023;389:137–47.
  5. Wu YL, Dziadziuszko R, Ahn JS, Barlesi F, Nishio M, Lee DH, et al. Alectinib in resected ALK-positive non-small-cell lung cancer. N Engl J Med. 2024;390:1265–76.
  6. Gemma A, Kusumoto M, Sakai F, Endo M, Kato T, Saito Y, et al. Real-world evaluation of factors for interstitial lung disease incidence and radiologic characteristics in patients with EGFR T790M-positive NSCLC treated with osimertinib in Japan. J Thorac Oncol. 2020;15:1893–906.

Comment #14: Address the clinical relevance of margin recurrence more directly.

Answer #14: We had already discussed its impact on survival, but we added the following sentence to further emphasize in the Discussion section that the high rate of local recurrence after partial resection is a significant concern.

Changes #14: (Discussion, line 256-259) Although wedge resection may be acceptable from a safety standpoint in patients with ILD, it has not been adopted as the standard surgical procedure due to lower survival rates caused by a high incidence of local recurrence, including recurrence at the surgical margins [31].

 (Reference)

  1. Ginsberg RJ, Rubinstein LV; Lung Cancer Study Group. Randomized trial of lobectomy versus limited resection for T1 N0 non-small cell lung cancer. Ann Thorac Surg. 1995;60(3):615–622. doi:10.1016/0003-4975(95)00537-U

 Comment #15: Highlight that the 8% AE-ILD rate in WG suggests risk is not eliminated, which should caution against overreliance on wedge resection.

Answer #15: Thank you for your thoughtful comment, we added the sentence in Conclusion section as below.

Changes #15: (Conclusion, line 286-287) However, it does not eliminate the risk of postoperative acute exacerbation of ILD and not show significant difference in mortality.

Conclusion

Comment #16: Add that the lack of standardized surgical protocols across institutions could introduce variability. (Suggestion)

Answer #16: Thank you for your thoughtful comment, we added the sentence in limitation as below.

Changes #16: (Discussion, line268-269) And surgical protocols across institutions could introduce variability.

Comment #17: Improve grammar and fluency in places (e.g., "The ILD group had high frequencies of these unfavorable characteristics” → “The ILD group showed a high prevalence of unfavorable characteristics…”).

Answer #17: Thank you for your kind comment, we made several conscious revisions in multiple sections.

Round 2

Reviewer 1 Report

Comments and Suggestions for Authors

In the manuscript, the authors present a study regarding the utility of the wedge resections in patients with lung cancer and previous interstitial lung disease. In my opinion, it is an interesting manuscript that can be published. The manuscript has been reviewed before and the authors changed the manuscript according to the previous reviewer indications. Their comments are quite pertinent. Now, the value of the manuscript has been improved. That is why, I think that this manuscript can be published in this form.